

**Influence of functional groups on toxicity of carbon nanomaterials:**
**implication for toxicological evolution during atmospheric relevant**
**aging of soot**
Yongchun Liu [1, 2], Haotian Jiang [2, 4], Chunmei Liu [3], Yanli Ge [2], Lian Wang [2], Bo Zhang [2],
Hong He [2, 4,5], Sijin Liu [2, 4]
[1] Aerosol and Haze Laboratory, Advanced Innovation Center for Soft Matter Science and
Engineering, Beijing University of Chemical Technology, Beijing, 100029, China
[2] State Key Joint Laboratory of Environment Simulation and Pollution Control, Research Center
for Eco-Environmental Sciences, Chinese Academy of Sciences, Beijing, 100085, China
[3] Bioduro Technology (Beijing) Co., Ltd., Beijing, 102200, China
[4] University of Chinese Academy of Sciences, Beijing, 100049, China
[5] Center for Excellence in Urban Atmospheric Environment, Institute of Urban Environment,
Chinese Academy of Sciences, Xiamen 361021, China.
*Correspondence to:* Y. Liu (liuyc@buct.edu.cn) and S. Liu (sjliu@rcees.ac.cn)
**Abstract:**
It has been well recognized that carbon nanomaterials and soot particles are toxic for
human health, while it is still controversial about the influence of functionalization on
their toxicity as well as the evolution of the toxicity of carbon nanomaterials due to
chemical aging in the atmosphere. In the current study, the oxidation potential measured
by dithiothreitol (DTT) decay rate and the cytotoxicity to murine macrophage cells of
different functionalized carbon nanomaterials were investigated for understanding the
role of functionalization in their toxicities. The DTT decay rates of special black 4A





(SB4A), graphene, graphene oxide, single wall carbon nanotubes (SWCNT), SWCNT-
OH and SWCNT-COOH were $45.9\pm3.0$, $58.5\pm6.6$, $160.7\pm21.7$, $38.9\pm8.9$, $57.0\pm7.2$ and
$36.7\pm0.2$ pmol $min^{-1}\mu g^{-1}$, respectively. Epoxide was found to be mainly responsible for
the largest DTT decay rate of graphene oxide compared with other carbon
nanomaterials based on comprehensive characterizations. Both carboxylation and
hydroxylation showed little influence on the oxidation potential of carbon
nanomaterials, while epoxidation contributes to the enhancement of oxidation potential.
All these carbon nanomaterials were toxic to murine J774 cell line. However, oxidized
carbon nanomaterials (graphene oxide, SWCNT-OH and SWCNT-COOH) showed
weaker cytotoxicity to J774 cell line compared with the corresponding control sample
as far as the metabolic activity was considered and stronger cytotoxicity to J774 cell
line regarding to the membrane integrity and DNA incorporation. These results imply
that epoxidation might enhance the oxidation potential of soot particles during transport.



## Introduction


Soot, which originates from incomplete combustion, is a mixture of elemental
carbon and organic carbon (OC) compounds (Muckenhuber and Grothe, 2006). The
adverse effect of soot particles on human health has attracted much attention in the
atmospheric chemistry community (Baumgartner et al., 2014). For example,
mitochondrial damage in alveolar macrophages and bronchial epithelial cells resulted
from exposure of diesel exhaust particles (DEPs) has been observed (Li et al., 2002a;Li
et al., 2002b). Oxidation stress or reactive oxygen generation (ROS) is one of
mechanisms related to the toxicity of particles including soot particles (Nel et al., 2006),
and has been even used as a paradigm to assess particle toxicity (Xia et al., 2006).
Dithiothreitol (DTT) decay rate is commonly used as a cell-free measure of the
oxidative potential of different particles (Cho et al., 2005;Charrier and Anastasio,
2012;Kumagai et al., 2002), such as ambient particles (Li et al., 2003;Fang et al.,
2016;Cho et al., 2005;Charrier and Anastasio, 2012;Wang et al., 2013), secondary
organic aerosol (SOA) (McWhinney et al., 2013b), DEP (Li et al., 2009;McWhinney et
al., 2013a), carbon nanotubes (CNT)(Liu et al., 2015), flame soot (Antinolo et al.,
2015;Holder et al., 2012;Li et al., 2013) and commercial carbon black (CB) particles
(Koike and Kobayashi, 2006;Li et al., 2009;Li et al., 2015;Li et al., 2013). However,
the reported DTT decay rate of soot and CB particles varied substantially, from 0.9 to
~50 pmol min$^{-1}$ μg$^{-1}$. The variation of DTT decay rate among different samples implies
the importance of the composition or structure of particles in their toxicities.
Although transition metals, element carbon, humic-like substances and quinones



are responsible for ROS generation on particle surface (McWhinney et al., 2013b;Li et
al., 2003), more work is still required to deeply understand the toxicity of soot and the
reason why the toxicity varies greatly among different soot samples. On the other hand,
soot particles are prone to undergo oxidation by $O_3$, OH and $NO_3$ etc. during transport
in the atmosphere. Subsequently, functionalization including formation of OH, C=O,
epoxide (C-O-C) and COOH occurs (Mawhinney et al., 2000;Liu et al., 2015;Holder et
al., 2012). This make it more complicate to understand the toxicity of soot particles.
For example, several studies have found that atmospheric relevant oxidation of CB or
BC by $O_3$ leads to enhancement of their oxidative potential (Li et al., 2009;Li et al.,
2013;Li et al., 2015;Antinolo et al., 2015;Holder et al., 2012). In particular, the DTT
decay rate of soot particles has been found increasing as a function of the content of
quinone formed via ozone oxidation of organic carbons in soot (Antinolo et al., 2015).
However, some other studies have found that oxidation of CB or soot by $O_3$ or OH
under atmospheric related conditions has little influence on their oxidative potential or
cytotoxicity although surface functionalization is observable (Liu et al., 2015;Peebles
et al., 2011). Therefore, it is necessary to understand the role of functional groups in the
toxicity of soot particles.
During combustion process, however, multiple functional groups including OH,
C=O, COOH, esters and so on are usually formed at the same time and present in both
OC and EC (Han et al., 2012a). Thus, it is difficult to differentiate the role of one kind
of functional group from others in the toxicity of soot particles. Carbon black (CB),
which is produced from incomplete combustion of heavy petroleum materials under



controlled conditions (Apicella et al., 2003), and engineered carbon nanomaterials are
a quasi-graphitic form of nearly pure element carbon (EC, consist of graphene layers)
and are distinguished by its very low quantities of extractable organic compounds and
total inorganics (Long et al., 2013) compared with soot. Therefore, it is possible to
investigate the role of functional groups in the toxicity of soot when using CB or
engineered carbon particles with different functional groups as model sample of soot
particles. Actually, it has been recognized that the surface properties of carbon
nanomaterials will influence their biological effects or toxicity (Lara-Martinez et al.,
2017;Liu et al., 2014b;Koromilas et al., 2014). For example, a recent study has found
that hydrated graphene oxide exhibited a higher cytotoxicity to THP-1 and BEAS-2B
cells as a consequence of lipid peroxidation of the surface membrane and membrane
lysis compared to pristine and reduced graphene oxide (Li et al., 2018). Functionalized
multiwalled carbon nanotubes (fMWCNTs) is highly cardioembryotoxic in comparison
with Functionalized oxygen-doped multiwalled carbon nanotubes (fCOxs) (Lara-
Martinez et al., 2017). As pointed out by Lara-Martinez et al. (2017), however,
cytotoxic effects of carbon nanomaterials at the cellular level generate considerable
controversy and more research is clearly needed to gain insight into the mechanism of
these adverse effects. In addition, passive diffusion and energy-dependent endocytosis
are the two methods suggested for particles entry into living cells. They can also be
distributed to various parts of the body, from where they can either remain, translocate,
or be excreted. Therefore, it is meaningful to investigate the influence of
functionalization on other endpoints alone even for these carbon nanomaterials.





In the current study, both the cell-free toxicity and the cell cytotoxicity of carbon
nanomaterials with different functionalities were evaluated to focus on the role of
functionalization in their toxicities to understand the possible influence of different
source or oxidation processes on the toxicity evolution of soot particles during transport
in the atmosphere. DTT decay rate representing the oxidative potential and the
cytotoxicity of murine macrophage cell were investigated. The carbon nanomaterials
were characterized with inductively coupled plasma-mass spectrometry (ICP-MS),
thermal gravity analysis (TGA), X-ray photoelectron spectroscopy (XPS), transmission
electron microscopy (TEM) and zeta potential analyzer. The role of oxygen containing
species in the toxicity of carbon nanomaterials was discussed. This work will be helpful
for understanding the toxicity evolution of soot during oxidation in the atmosphere and
evaluation the toxicity of engineered nano-particles.
**Experimental Section**
**Chemicals and characterization of particle samples.** Commercial carbon
nanomaterials including Special Black 4A (SB4A), graphene, graphene oxide, SWCNT,
SWCNT-OH and SWCNT-COOH were used in this study. All these functional groups
have been identified in soot particles and chemical aged soot or CB particles. SB4A
was supplied by Degussa. The other carbon nanomaterials with purity >98% were
supplied by Timesnano. To obtain graphene oxide with low epoxide content, graphene
oxide were thermally treated at 200 °C for 30 min in high purity (99.999%) nitrogen
flow. Dithiothreitol (DTT) was supplied by Sigma-Aldrich. 5,5′-dithiobis-(2-
nitrobenzoic acid) (DTNB) was obtained from Alfa Aesar. Standard solutions of metal

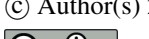



ions including Cr, Mn, Fe, Co, Ni, Cu, Zn, Cd, As, Sn and Pb were supplied by National
Institute of Metrology, China. 30 % $H_2O_2$ solution was supplied by Sinopharm
Chemical Reagent Co., Ltd.
A transmission electron microscope (H-7500, Hitachi) was used to investigate the
morphologies of carbon nanomaterials. Particles were ultrasonically dispersed in
ultrapure water (18 MΩ) and a droplet of suspending liquid was deposited onto a Cu
microgrid. An acceleration voltage of 80 kV was used for measurements. The
morphologies were shown in Fig. S1. The diameter of primary particles were analyzed
by ImageJ 1.41 software (Liu et al., 2010). The diameter of the primary carbon sphere
for SB4A was 66±17 nm SB4A. The out diameter (OD) of SWCNT, SWCNT-OH and
SWCNT-COOH was <2 nm with fiber length of 1-3 μm according to the product report
and also confirmed by TEM (Fig. S1). Graphene and graphene oxide were 2-
dimensional materials with monolayer and the diameter of 0.5-3 μm.
XPS were measured using an AXIS Supra/Ultra (Kratos, Kratos Analytical Ltd.) to
identify the oxygen containing species on the surface of carbon nanomaterials. The
samples were excited by Al Kα X-ray ($hv$=1486.7 eV) with 15 kV of working voltage
and 40 mA of emission current. The spectra were analyzed with XPS Peak software.
The content of organic carbon (OC) in carbon nanomaterials was measured by thermal
desorption using a commercial TG instrument (TGA/DSC1/HT1600, Mettler-Toledo
Co., Ltd.). The amount of OC lost from the particles was recorded when the temperature
was ramped from 30 to 300 °C at 10 °C min$^{-1}$ in nitrogen flow according to the protocol
reported in previous work (Han et al., 2012a). Metals in the particles were measured




with an inductively coupled plasma mass spectrometer (ICP-MS 7500a, Agilent
Technologies) after digested with concentrated 1:3 $HNO_3$/HCl. Transition metals were
quantified with the standard solution. Zeta potentials of the carbon nanomaterials were
measured after sonicating 30 min in ultrapure water (18.2 M$\Omega$) by using a Nanoparticle
Size & Zeta Potential Analyzer (Zetasizer Nano, ZS90).
**DTT assay test.** The DTT assay is an indirect chemical assay used for measuring the
redox cycling capacity of PM. The added DTT is oxidized to its disulfide form by the
ROS in particulate matter (Kumagai et al., 2002). Thus, the rate of DTT consumption
is proportional to the concentration of the ROS in the sample (Cho et al., 2005). In this
study, ~150 μg carbon nanomaterials were suspended in 10.0 ml phosphate buffer (0.1
M, pH 7.4) and sonicated for 15 min. 2.0 ml of 0.5 mM DTT solution was added to 3.0
ml aliquots of the sonicated suspensions. A redox reaction took place in a thermostat
shaking chamber at 37 ºC. The remained DTT concentration was measured every 15
minutes by adding 0.25 ml of the reaction mixture filtration to 1.0 ml of 0.25 mM
DTNB solution. DTNB reacted with the thiol groups in DTT to form a yellow
compound (2-nitro-5-thiobenzoate, NTB), which could be detected by UV-vis
absorption spectrometer (723N, Shanghai Ruiting Technology Co., Ltd) at 412 nm.
Then, the amount of DTT consumed by PM was calculated according to the standard
curves of DTT. The loss rate of DTT via a redox reaction in the presence of PM was
monitored as the concentration decrease of DTT and normalized to the particle mass.
Blank experiments were carried out without carbon nanomaterial particles in the buffer
solution. For some samples, the response to the DTT assay was also measured for the



water soluble components of SWCNT by filtering aliquots of the samples with a 0.22
μm syringe PTFE filter, and measuring the activity of the solution without particles.
**In vitro assays**. Carbon nanomaterial particles were dispersed with 0.025% Tween-80
in 0.19% NaCl solution using a Dounce glass homogenizer, followed by sonication. A
homogeneous and stable suspension of SWCNTs was obtained after the sonication
process. Cytotoxicity assessment of carbon nanomaterials was carried out using the
murine J774 cells. Three different assays targeting distinct mechanisms of cellular
metabolic perturbations were assessed simultaneously, including ATP (energy
metabolism), LDH (membrane integrity) and BrdU (incorporation into DNA) assays.
The experiments were carried out according to the corresponding protocol. Briefly, $4 \times$
$10^5$ J774 cells ml$^{-1}$ were exposed to carbon particles in 96-well plates for 24 hours for
ATP and LDH assays, while the initial J774 cell concentration was $2 \times 10^5$ cells ml$^{-1}$
for BrdU assay. Carbon nanomaterials were dosed at 0, 10, 30 and 100 μg cm$^{-2}$ in a
final volume of 200 μl well$^{-1}$ as similar to that reported in literatures (Kumarathasan et
al., 2014;Kumarathasan et al., 2012). The luminescence spectroscopy of the supernatant
after centrifugal separation at 1000 rpm for 5 min was measured after 24 h of cell
exposure using a Multimode Microplate Reader (Varioskan®Flash, Thermo Fisher
Scientific). The zero dose of carbon nanomaterials referred to the blank experiment and
also means the toxicity of 0.025% Tween-80 alone in 0.19% NaCl solution. Similar to
the literature results (Hadrup et al., 2017), they did not incur any obvious deleterious
effect on cells growth. In addition, it has been well recognized that carbon nano-
particles tended to aggregate in water even after ultrasonic dispersion. Tween-80 has





been verified to be a biocompatible dispersant for carbon black (Kim et al., 2012).
Negative control experiments were performed in wells containing medium without cells
to obtain a value for background luminescence. Positive control experiments were
carried out with $H_2O_2$ solution for LDH assays (Fig. S2).
**Results and discussion**
**Oxidative potential of carbon nanomaterials.** Figure 1 shows the DTT decay rates of
SB4A, graphene, graphene oxide, SWCNT, SWCNT-OH and SWCNT-COOH. They
were 45.9±3.0, 58.5±6.6, 160.7.0±21.7, 38.9±8.9, 57.0±7.2 and 36.7±0.2 pmol min$^{-1}$μg$^{-1}$
$^{1}$, respectively. Except for graphene oxide, the measured DTT decay rates for these
carbon nanomaterials (with mean value of 47.4±10.1 pmol min$^{-1}$μg$^{-1}$) were comparable
with the DTT loss rates of BC reported in the literatures. For example, it was 36.2±4.9
pmol min$^{-1}$ μg$^{-1}$ for Printex U (Li et al., 2015) and 59.3±7.4 pmol min$^{-1}$ μg$^{-1}$ for SWCNT
(Liu et al., 2015). These values were also comparable with that of the typical soot
particles (BC), such as 33.6 pmol min$^{-1}$ μg$^{-1}$ for methane flame soot (Holder et al., 2012),
49±7 pmol min$^{-1}$ μg$^{-1}$ for propane flame soot (Antinolo et al., 2015), 27.0 pmol min$^{-1}$
μg$^{-1}$ for hexane flame soot (Li et al., 2013), as well as the typical ambient $PM_{2.5}$ particles
(34.7±19.1 pmol min$^{-1}$ μg$^{-1}$) (Charrier and Anastasio, 2012;Liu et al., 2014a). However,
the measured DTT decay rates for these carbon nanomaterials were significantly higher
than that of diesel soot (6.1 pmol min$^{-1}$ μg$^{-1}$) and graphite (0.9 pmol min$^{-1}$ μg$^{-1}$) (Li et
al., 2013) reported in previous work. It should be noted that the DTT decay rate of
graphene oxide measured in this study was 160.7±21.7 pmol min$^{-1}$ μg$^{-1}$. Based on T-
test, the DTT decay rate of graphene oxide was significantly higher than that of other





tested carbon nanomaterials at the 0.05 level ($t$=8.498, which is greater than the critical
value of 2.447). This means that graphene oxide definitely has a stronger oxidative
potential than other CB or carbon nanomaterials in this work.
**Cytotoxicity of carbon nanomaterials to murine J774 cell line.**

At the present time, the A549 (a human adenocarcinomia alveolar epithelial cell)

and THP-1 (a human leukemia monocytic cell line) cell lines were usually chosen as
target cell lines (Kumarathasan et al., 2012;Kumarathasan et al., 2014;Liu et al., 2015)
to evaluate the alveolar and pulmonary toxicity of CB particles. As the first barrier of
the immune system, macrophage cell lines will fight against the invaded particles in the
lungs. Macrophage cell lines like J774 cells are ideal model systems for establishing
the biophysical foundations of autonomous deformation and motility of immune cells
(Lam et al., 2009). It has been found that CB nanoparticles are able to stimulate the
release of macrophage chemo-attractants when exposed to type II epithelial cell lines
(L-2 cells) at sub-toxic doses (Barlow et al., 2005). CNTs exposure can also lead to
biological changes in J774 cells (Kumarathasan et al., 2012). Therefore, it is meaningful
to investigate the cytotoxicity of different carbon nanomaterials as well as the influence
of surface functional group on the macrophage cell lines.

Figure 2 shows the in vitro toxicities of SB4A, graphene, graphene oxide, SWCNT,

SWCNT-COOH and SWCNT-OH. The stars mean the indicator of the toxicity at a
certain dose of carbon nanomaterials is significantly different from the corresponding
blank experiments at 0.05 level. As shown in Fig. 2, the metabolic activity of J774 cell
line decreased monotonously as a function of the dose of all these carbon nanomaterials.



This means the carbon nanomaterials investigated in this work are toxic to murine J774
cell line. This is consistent with the previous results that CNT and Printex U are toxic
to J774 cells (Kumarathasan et al., 2012) and graphene oxide can induce dose-
dependent cell death in normal lung fibroblasts (HLF), macrophages (THP-1 and
J744A), epithelial (BEAS-2B) cells, lung cancer cells A549 etc. (Zhang et al., 2016;Li
et al., 2018).

In Fig. 2A, the relative ATP level (1.01±0.02) at the SB4A dose of 10 μg cm$^{-2}$ was

almost the same as that of the blank sample, while it significantly decreased to
0.89±0.05 and 0.61±0.07 when the dose of SB4A increased to 30 μg cm$^{-2}$ and 100 μg
cm$^{-2}$, respectively. Similarly, the relative ratio of BrdU incorporation decreased from
0.74±0.03 to 0.60±0.04 when the dose of SB4A increased from 30 to 100 μg cm$^{-2}$. This
means SB4A is also an inhibitor for cell proliferation of murine J744. However, the
released LDH levels were constant within experiment uncertainty at different SB4A
doses. This means the cell membrane might be intact when exposed to SB4A.

As shown in Fig. 2B-F, the metabolic activity of murine J774 cell decreased more

significantly when exposed to engineered carbon nanomaterials than SB4A. For
example, the relative ratio of ATP level was 0.67±0.06, 0.84±0.03, 0.59±0.10,
0.93±0.01 and 0.88±0.02 even when the J774 cells were exposed to 10 μg cm$^{-2}$
graphene, graphene oxide, SWCNT, SWCNT-OH and SWCNT-COOH, respectively.
When exposed to high doses of engineered carbon nanomaterials, the reduction of
relative ATP level became more significant. These results mean the cytotoxicity of the
engineered carbon nanomaterials studied in this work are stronger than that of SB4A





regarding to metabolic activity. Graphene, graphene oxide and SWCNT-COOH
significantly enhanced release of LDH at different exposure levels, while SWCNT and
SWCNT-OH only led to significant increases of released LDH at high exposure level
(100 μg cm$^{-2}$). This implies the integrity of cell membrane decreased when J774 cells
were exposed to these engineered carbon nanomaterials. This might be related to lipid
peroxidation induced by these particles (Li et al., 2018).
It should be noted that the reduction of ATP ratio of J774 cells exposed to graphene
oxide was weaker than that of graphene. The reduction of ATP ratio of J774 cells
exposed to SWCNT-OH or SWCNT-COOH was also weaker than that of SWCNT.
However, compared with graphene, graphene oxide showed much stronger toxicity to
J774 cell as far as the membrane integrity was considered. The released LDH at
exposure level of 30 μg cm$^{-2}$ graphene oxide was comparable with that when exposed
to 150 ppm $H_2O_2$ (Fig. S2). In addition, graphene oxide, SWCNT-OH and SWCNT-
COOH significantly inhibited DNA synthesis of J774 cells when the carbon
nanomaterials doses were above 10 μg cm$^{-2}$, while graphene and SWCNT did not show
significant inhibition of DNA synthesis for J774 cells. For instance, the relative ratio of
BrdU when J774 cells exposed to 100 μg cm$^{-2}$ of graphene oxide was 0.61±0.10, while
it was 0.77±0.10 for graphene exposed cells at the same exposure level. They were
0.62±0.10 for SWCNT-OH and 0.56±0.09 for SWCNT-COOH treated cell at a dose of
10 μg cm$^{-2}$ compared with 0.83±0.09 for 10 μg cm$^{-2}$ of SWCNT treated J774 cell. These
results suggested that functionalized carbon nanomaterials caused a low cytotoxicity of
murine J774 cell line regarding to the cell apoptosis, while a stronger toxicity was



demonstrated for cell proliferation and the membrane integrity. This finding was true,
in particular, for graphene oxide.
**Influence of physiochemical properties on the toxicity of different samples.** It
should be pointed out that the morphologies of these carbon nanomaterials varied
greatly. SB4A was a zero dimensional material. SWCNT, SWCNT-OH and SWCNT-
COOH were one dimensional materials. Graphene and graphene oxide were two
dimensional materials (Fig. S1). The DTT decay rate (Fig. 1) did not show obvious
dependence on their morphologies in this work. For example, except for graphene oxide,
the DTT decay rates were comparable among all the other materials regardless of the
morphology. Graphene and graphene oxide showed similar particle size, graphene layer
and morphologies (Fig. S1), while they showed totally different toxicity as shown in
Fig. 1. In Fig. 2, the cytotoxicity of SB4A, graphene and SWCNT showed an increase
trend regarding the metabolic activity of J774 cell. This can be explained by the
different mode of action (MOA) when the cells were exposed to different types of
nanomaterials. For example, adhesions and/or covering on cells could be the main
MOA for graphene/graphene oxide (2-D structure), while for carbon nanotubes (1-D
structure), piercing and/or internalization by cells could be the main MOA. This means
morphology should plays a role in determining the cytotoxicity of the carbon
nanomaterials studied in this work. Therefore, in the following section we mainly
discuss the cytotoxicity among these materials having same dimension, such as
SWCNT-OH and SWCNT-COOH verse SWCNT and graphene oxide verse graphene.
In addition, as shown in Fig. S3, all these carbon nanomaterials revealed negative zeta





301 potential from -42 mV to -20 mV. SB4A, graphene oxide and SWCNT-COOH almost

302 borne the same zeta potential (-42 mV), while SWCNT, SWCNT-OH and graphene

303 showed comparable zeta potential. This observation suggested the stability of dispersed

304 SB4A, graphene oxide and SWCNT-COOH in water and the interaction between these

305 particles with cells was comparable.

306  Transition metals in the particles have been identified to be the important

307 contributor to ROS generation (McWhinney et al., 2013b;Li et al., 2003). The content

308 of transition metals including Cr, Fe, Mn, Co, Ni, Cu, Zn, As, Cd, Sn and Pb were

309 measured by using an ICP-MS after the carbon nanomaterials were digested with 1:3

310 $HNO_3$/HCl. As shown in Fig. S4A, Fe was the most abundant transition metal in these

311 carbon nanomaterials. Its concentration varied from 122 $\mu g\ g^{-1}$ to 6596 $\mu g\ g^{-1}$ among

312 different carbon nanomaterials. The concentration of other metals varied from zero to

313 several hundred $\mu g\ g^{-1}$ depending on both carbon nanomaterials and the type of metals.

314 Compared with SB4A, these engineered carbon nanomaterials showed higher metal

315 content. For example, the total metal content in graphene was 6 times as high as that in

316 SB4A, while it was 33 times in SWCNT as high as that in SB4A. This can be explained

317 by the fact that graphene and SWCNT materials were catalytically synthetized using

318 metal catalysts containing Fe, Co or Ni. It should be noted that although the metal

319 content of SB4A was very low compared with other materials, the DTT decay rate of

320 SB4A was still comparable with these engineered carbon nanomaterials except for

321 graphene oxide as shown in Fig. 1. On the other hand, SWCNT had the highest metal

322 content, while graphene oxide rather than SWCNT showed the strongest DTT decay





rate. In addition, the soluble metal contents were in the following order: SWCNT-
COOH > SWCNT > SB4A > graphene oxide > graphene > SWCNT-OH (Fig. S4B),
after being sonicated for 30 min in water. Graphene oxide (103.7 µg g$^{-1}$) did not show
a significant difference compared with SB4A (106.3 µg g$^{-1}$) and graphene (93.7µg g$^{-1}$).
These results indicated that the high oxidative potential of graphene oxide relative to
other materials cannot be attributed to their difference in bounded or soluble transition
metals. This can be explained by the following reasons. First, metal content were
measured after digested with 1:3 HNO$_3$/HCl. The speciation of metals should be quite
different from that presenting in the pristine carbon nanomaterials. For example, the
contents of soluble metal ions after sonicated for 30 min (Fig. S4B) varied from zero to
356 µg g$^{-1}$. These values were much lower than the corresponding metal contents of
digested samples as shown in Fig. S4A. Second, metal might be in the inner pores of
carbon nanomaterials. This will decrease the efficiency of metals to generate ROS.
Finally, the concentration of carbon nanomaterials was 10-40 µg ml$^{-1}$ in DTT assay tests.
This meant the concentration of transition metals was at ng ml$^{-1}$ level even if all of the
transition metals were available. The low concentration of metals released might lead
to negligible contribution to ROS formation. This was further confirmed by the very
small DTT decay rate of the SWCNT filtered solution (1.66±0.15 pmol min$^{-1}$ µg$^{-1}$)
compared with that of SWCNT suspension (38.9±8.9 pmol min$^{-1}$ µg$^{-1}$) even though
SWCNT had the highest metal concentration (Fig. S4A). This was consistent with the
previous conclusions that redox activity originates from the particle surface of CB or
BC materials but not from water-soluble substances (Liu et al., 2015;McWhinney et al.,




2013a).

Figure 3 shows the thermo gravity and differential thermal analysis curves for these

CB materials when the temperature was ramped from 30 to 300 °C at 10 °C min$^{-1}$ in
nitrogen flow. Weight loss (Fig.3A) accompanied with an endothermic process (Fig. 3B)
were observed below 60°C for all of these samples. This can be ascribed to desorption
of surface adsorbents including organics and trace water. As shown in Fig. 3B, the
saddle points of these differential thermal analysis curves were observed at 35, 35, 41,
42, 56 and 58 °C for graphene, SWCNT, SB4A, SWCNT-OH, SWCNT-COOH and
graphene oxide, respectively. It should be noted that the oxidized carbon nanomaterials
such as SWCNT-OH, SWCNT-COOH and graphene oxide showed higher saddle points
of the heat curves than graphene, SWCNT and SB4A. This implies stronger interaction
between the adsorbents and these three oxidized carbon nanomaterials compared with
the counterpart. Therefore, it is reasonable to deduce that the adsorbed water mainly
contribute to the weight loss in this stage. The sample weight slightly decreased as the
temperature further increased for all of these carbon nanomaterials except for graphene
oxide and accompanied with a gradual increase of the heat flow. This can be ascribed
to desorption of adsorbed organics from the surface of the carbon nanomaterials. The
relative small increase rate of the heat in this stage was consistent with the small heat
capacity of organics when compared with the first one which was ascribed to desorption
of water. For graphene oxide, however, weight loss (from 32% to 60%) was
significantly observed accompanied with an acute exdothermic process when the
temperature increased from 150 to 200 °C as shown in Fig. 3B. This implies that release



of pyrolysis products and structure collapse of graphene oxide occur. It also means a
high reactivity of graphene oxide and highlights the distinctive property of graphene
oxide among these investigated carbon nanomaterials.

The adsorbed organics were estimated based on the thermogravimetric curves when

the possible contribution of water was ruled out. For graphene oxide, 150 °C was taken
as the endpoint, while 300 °C was chosen for other samples. The content of adsorbed
organics on SB4A, graphene, graphene oxide, SWCNT, SWCNT-OH and SWCNT-
COOH was 6 %, 13 %, 15 %, 9 %, 5 % and 9 %, respectively, as shown in the insert
graph of Fig. 3A. The content of organics cannot explain the sequence of the DTT loss
rate (Fig. 1) and the cytotoxicity (Fig. 2) of these carbon nanomaterials. For example,
the content of organics on graphene and graphene oxide were almost the same, while
the DTT decay rate of graphene oxide was as about 2.5 times as that of graphene (Fig.
1) and the cytotoxicity of graphene oxide as for metabolic activity to murine J774 was
weaker than that of graphene. In addition, the organic content of SWCNT was the same
as that of SWCNT-COOH, while SWCNT-COOH showed weaker toxicity to murine
J774 cell line than SWCNT as far as the metabolic activity was considered (Fig. 2).
This means the different toxicities observed in this study cannot be explained by the
adsorbed organics among these materials.

To further investigate the role of surface oxygen in the toxicity of carbon

nanomaterials, the oxygen-containing species of these carbon nanomaterials were
identified with X-ray photoelectron spectroscopy. Fig. 4 shows the typical O1s and C1s
spectra of these carbon nanomaterials. Adsorbed oxygen at 535.2 eV, carbon-oxygen





single bond in hydroxyl group (C-OH) at 533.5 eV, carbon-oxygen single bond in
epoxide (C-O-C) at 532.6 eV, carbon-oxygen double bound (C=O) at 531.8 eV and
highly conjugated form of carbonyl oxygen such as quinone groups at 530.5 eV
(Schuster et al., 2011) presenting in these CB samples as shown in Fig. 4A-F. In the
C1s spectra (Fig. 4G-L), the band at 291 eV was attributed to the shakeup peak
associated with $\pi$-$\pi$* transition (Simmons et al., 2006). The band at 289 eV
corresponded to carbonyls and epoxides was observed at 287 eV (Kuznetsova et al.,
2001). The band at 285 eV and 284.6 eV was assigned to graphite and $sp^3$ carbon,
respectively. In particular, the intensity of C-O-C at 532.6 eV in graphene oxide was
very strong compared with other carbon nanomaterials. At the same time, the band of
C-O-C at 287 eV was also much stronger than that of other carbon nanomaterials in the
C1s spectrum. These results mean that epoxides (C-O-C) is the predominate species
(Fig. 5C and I) in graphene oxide.
Fig. 5A summarizes the distribution of the oxygen species mentioned above
normalized to O atoms in these carbon nanomaterials. Highly conjugated form of
carbonyl oxygen (quinone) and adsorbed oxygen contributed little to the total oxygen
on the surface (<1 %), while C=O, C-O-C and C-OH were predominate oxygen-
containing species. Our results agree well with the previous work that C=O, C-O-C and
C-OH dominated oxygen-containing species on natural chars, diesel soot, hexane soot
and activated charcoal (Langley et al., 2006). Although quinone has been well
recognized to contribute to ROS generation on the surface of fine particles (Kumagai
et al., 2002;Li et al., 2002b), the content of quinone was lower than 0.35% and showed





little difference among all of these tested carbon nanomaterials (Fig. 5A and B). It did
so for adsorbed oxygen content. Therefore, we can conclude that the very large DTT
decay rates of graphene oxide compared with other carbon nanomaterials as shown in
Fig. 5C cannot be explained by the content of quinone or adsorbed oxygen.
As shown in Fig. 5A, the total oxygen content of SB4A, graphene, SWCNT,
SWCNT-OH and SWCNT-COOH was 6.68%, 2.41 %, 2.88%, 3.60% and 9.21%,
respectively. They were comparable with that of diesel soot (2.1%-12.2%) (Schuster et
al., 2011). However, the oxygen content of graphene oxide (29.0%) was significantly
higher than the other carbon nanomaterials (Fig. 5A). At the same time, the distribution
pattern of the surface species on graphene oxide was quite different from the other
carbon nanomaterials. Fig. 5B compared the content of the oxygen-containing species
of graphene oxide with other carbon nanomaterials. The red stars indicate the content
of oxygen-containing species in graphene oxide, while the blue boxes show that of other
carbon nanomaterials. It can be seen that the content of quinone and adsorbed oxygen
showed no difference between graphene oxide and other carbon nanomaterials. The
concentration of C=O and C-OH in graphene oxide was slightly higher than that in the
other carbon nanomaterials. However, the content of epoxide in graphene oxide was
significantly higher than the other carbon nanomaterials. The content of epoxide in
graphene oxide normalized to O atoms was 20.8 %, which was 71.7 % of its total
oxygen content (Fig. 5B), while it was less than 2.7 % in other carbon nanomaterials.
This well corresponded to the large DTT decay rates of graphene oxide (160.7 pmol
min$^{-1}$ μg$^{-1}$) compared to other carbon nanomaterials (less than 60 pmol min$^{-1}$ μg$^{-1}$) as




shown in Fig. 5C. It should be noted that the content of epoxide was not linearly
correlated to the DTT activity. This can be explained by the typical nonlinear
relationship between the dose of toxicant and toxicity (Antinolo et al., 2015). It should
be pointed out that multiple parameters of particle may have influence on its toxicity,
in particular, on the cytotoxicity. For example, particle size and morphology may have
influence on the material mobility and uptake by cells. However, the above results at
least imply that these physiochemical properties such as morphology, metal and OC
content should not be crucial factors as for the toxicity of these carbon nanomaterials
because it is difficult to observe an obvious dependence of the toxicity on these factors.
In the meantime, we can propose that epoxides in graphene oxide are mainly
responsible for the high ROS activity of graphene oxide. The high ROS formation
potential of graphene oxide might also explain its strong cytotoxicity to J774 cell line
regarding to the cell membrane.

To further confirm this assumption, we measured the ROS activity of the thermally

treated graphene oxide at 200 °C in nitrogen flow because C-O-C (epoxide) structure
can be broken under this condition as shown in Fig. 3 and discussed above. XPS spectra
confirmed the broken of epoxide by the fact that the content of epoxide in thermally
treated graphene oxide decreased to 4.3% from 20.9% in graphene oxide as shown in
Figs. S5 and S6. In addition, TEM results also showed that graphene oxide broke into
small sheets, whose morphology and particle size were close to that of SB4A and
graphene oxide or graphene (Fig. S1). At the same time, the DTT decay rate of the
thermally treated graphene oxide decreased to $54.9 \pm 9.8$ pmol min$^{-1}$ μg$^{-1}$ (Fig. 6). This





value was comparable to the DTT decay rates of other carbon nanomaterials, in
particular, graphene ($58.5\pm6.6$ pmol min$^{-1}$ µg$^{-1}$) (Fig. 1), while it was significantly lower
than the graphene oxide ($160.7.0\pm21.7$ pmol min$^{-1}$ µg$^{-1}$) as shown in Fig. 6. It should
be noted the total oxygen contents of thermally treated graphene oxide was 19.3 %,
which was lower than that of graphene oxide (29.0 %) but significantly higher than that
of other carbon nanomaterials. However, the DTT decay rate of thermally treated
graphene oxide was still comparable with other carbon nanomaterials. This further
highlights the importance of functional group in the toxicity. Therefore, it means that
epoxides in graphene oxide are the highly reactive site for ROS formation on the surface
of graphene oxide. This is for the first time to observe that epoxide is a highly reactive
site for ROS formation besides quinone on carbon nanomaterials. This result is also
well consistent with the previous founding that epoxides in graphene oxide can oxidize
$SO_2$ to sulfate (He and He, 2016).

However, we did not observed significant dependence of cytotoxicity to murine

J774 cell line and the content of oxygen-containing species on the surface of carbon
nanomaterials although oxidized CB materials showed reduced toxicity to J774 cell
lines as far as metabolic activity was considered. In particular, the difference in surface
oxygen content between graphene oxide and graphene was much higher than that
between SWCNT-OH/SWCNT-COOH and SWCNT (Fig. 5A), while the differences in
metabolic activity to J774 cell line between graphene oxide and graphene was similar
to that between SWCNT-OH/SWCNT-COOH and SWCNT. The pathways of cellular
toxicity induced by particles reside in both oxidative stress (ROS) and non-oxidative



stress dependent (Shvedova et al., 2012). Oxidative stress leads to selective oxidation
of mitochondrial CL, NADPH oxidase activation and MPO activation in neutrophils,
while non-oxidative stress results from interference with mitotic spindle and actin
cytoskeleton, and steric hindrance of ion channels. The interaction between target cells
and particles should be much complicated than that between DTT and particles. As
discussed above, the cytotoxicity of nano-particles relied on not only the mode of action
but also the chemical nature of particles. Therefore, the different responses of the
oxidation potential and the cytotoxicity to the epoxide content in these carbon materials
might be accounted for by different mechanisms of toxicity among these assays.
**Conclusion ad atmospheric implications**
The DTT decay rates of special black 4A (SB4A), graphene, graphene oxide, single
wall carbon nanotubes (SWCNT), SWCNT-OH and SWCNT-COOH were 45.9±3.0,
58.5±6.6, 160.7±21.7, 38.9±8.9, 57.0±7.2 and 36.7±0.2 pmol min$^{-1}$μg$^{-1}$, respectively.
Epoxide has been for the first time identified as a highly active functional group in the
carbon nanomaterials as far as the oxidation potential is considered.
Oxidation is a useful method to obtain functionalized CB materials with distinctive
performance in industry. It is also a primary process in the atmosphere relating to
chemical aging of particles including soot and CB particles. This process unusually
leads to formation of carbonyls, hydroxyls, carboxylic acids, esters, ethers and epoxides
on the surface of CB or BC particles. Previous work have found that oxidation of carbon
nanomaterials (SWCNT) by O$_3$ or OH under atmospheric related conditions has little
influence on their oxidative potential or cytotoxicity although carbonyls, carboxylic



acids and esters were formed (Liu et al., 2015). Similarly, surface functionalization was
observed for commercial CB materials by ozone oxidation, while increase in the
cytotoxicity of murine macrophages and release of inflammation markers upon
exposure to the oxidized CB were not observed (Peebles et al., 2011). However, some
other studies observed that oxidation process enhanced the oxidation potential (Li et al.,
2015;Li et al., 2013;Antinolo et al., 2015) as well as the cytotoxicity (Holder et al.,
2012) of CB and BC particles. Using the model carbon nanomaterials with different
dominate surface functionalities in this work, we have found that hydroxyl and carboxyl
functionalized CB particles had little influence on their oxidation potential, while
epoxide functionalized CB (graphene oxide) led to a very strong oxidation potential.
Epoxide has been identified as a surface product on SWCNT when treated with high
concentration of ozone (Mawhinney et al., 2000;Yim and Johnson, 2009). Besides
carboxylic acids, esters (Liu et al., 2015), ketone, lactone and anhydride species (Liu et
al., 2010;Han et al., 2012b), epoxides has also been identified as the surface product
during oxidation of SWCNT in atmosphere relevant conditions (Liu et al., 2015). This
means that oxidation potential enhancement of CB particles is also possibly resulted
from the formation of epoxide during chemical aging in the atmosphere. On the other
hand, graphene oxide was an important commercial product, while showed strong
oxidation potential as observed in this work. Therefore, Mussel-inspired chemistry is
necessary for fabrication of functional materials and decreasing their toxicity and for
biomedical applications (Liu et al., 2014b;Zhang et al., 2012).

It has been found that CB particles (Printex 90) can induce opening of plasma



membrane calcium channels leading to a calcium influx and cause significant release
of proinflammatory cytokine TNF-α by the murine J774 cells (M. et al., 2004),
subsequently potentially induce migration of macrophages (Barlow et al., 2005). This
could initiate the recruitment of inflammatory cells to sites of particle deposition and
the subsequent removal of the particles by macrophages. The metabolic activity of these
hydroxyl, carboxylic acid and epoxide functionalized carbon nanomaterials increased
when compared with the corresponding sample as observed in this work. This implies
chemical aging of these carbon nanomaterials might not pose an enhanced cytotoxicity
risk to macrophages although the oxidized carbon nanomaterials were still toxic as far
as metabolic activity was considered. However, the oxidized carbon nanomaterials
might pose enhanced cytotoxicity to macrophages regarding to membrane integrity and
DNA synthesis. It should be pointed out that exposure experiments were performed
under high particle concentration with short exposure time in this work. More work
needs to be done at low particle concentration with long exposure time in the future.
On the other hand, it has been found that aging rate of BC particles under highly
polluted urban environment is faster than that under clean conditions (Peng et al., 2016).
In the future, much work should be performed on the toxicity evolution of CB or BC
particles under real atmospheric conditions. Finally, it should be noted that the
interaction between particles and biological entities such as proteins or cells has not
been considered in this work. Therefore, the in vivo toxicological effect of these
functionalized particles needs to be further evaluated in the future.
**AUTHOR INFORMATION**





Corresponding Author
*E-mail: liuyc@buct.edu.cn, phone: +86-10-68471480, fax: +86-10-68471480 or
sjliu@rcees.ac.cn,

**AUTHOR CONTRIBUTION**
Y. L., H. H. and S. L. designed the experiments. Y. L. wrote the paper. Y. L., H. J. and
Y. G. did the DTT assay tests. C. L. and L. W. did the cytotoxicity assessments. H. J.
and B. Z. performed the characterization of samples.

**ACKNOWLEDGMENTS**
This research was financially supported by the National Natural Science Foundation of
China (91543109). YCL should thank Beijing University of Chemical Technology for
financial supporting.

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



**Figure captions**

**Figure 1**. DTT decay rates of several black carbon materials compared with literature

results (Li et al., 2013;Charrier and Anastasio, 2012;Liu et al., 2014a;Li et al., 2015;Liu

et al., 2015;Holder et al., 2012;Antinolo et al., 2015).

**Figure 2**. Cytotoxicity of (A) SB4A, (B) graphene, (C) graphene oxide, (D) SWCNT,

(E) SWCNT-OH and (F) SWCNT-COOH toward murine J774 cell line. The stars mean

the difference is significant at 0.05 level for a certain dose of carbon nanomaterials

compared with the corresponding blank experiments.

**Figure 3**. (A) Thermo gravity curves of carbon nanomaterials in nitrogen gas flow;

(B) the corresponding differential thermal analysis curves. The insert graph shows the

weight loss due to desorption of organics.

**Figure 4**. XPS spectra of carbon nanomaterials. (A)-(F) are O1s spectra and (G)-(L)

are C1s spectra for SB4A, graphene, graphene oxide, SWCNT, SWCNT-OH and

SWCNT-COOH, respectively.

**Figure 5**. (A) Distribution of oxygen containing species on the tested carbon

nanomaterials; (B) comparison of oxygen-containing species and (C) DTT decay rate

between graphene oxide and other carbon nanomaterials.

**Figure 6.** DTT decay rate for graphene oxide and thermally treated graphene oxide in $N_2$ flow

at 200 °C.



**Figures**

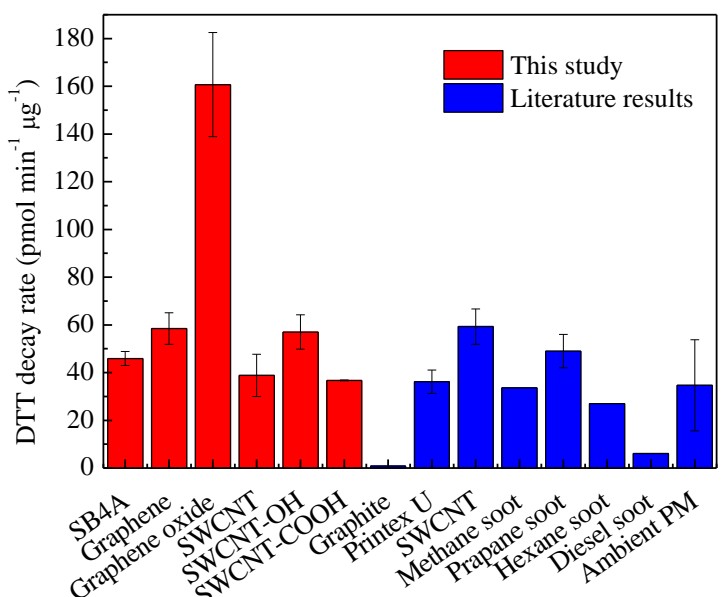


Fig. 1

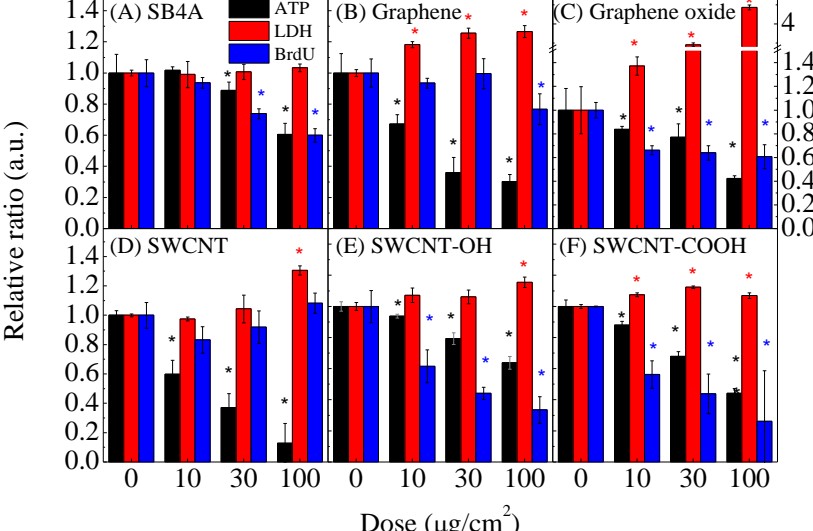


Fig. 2



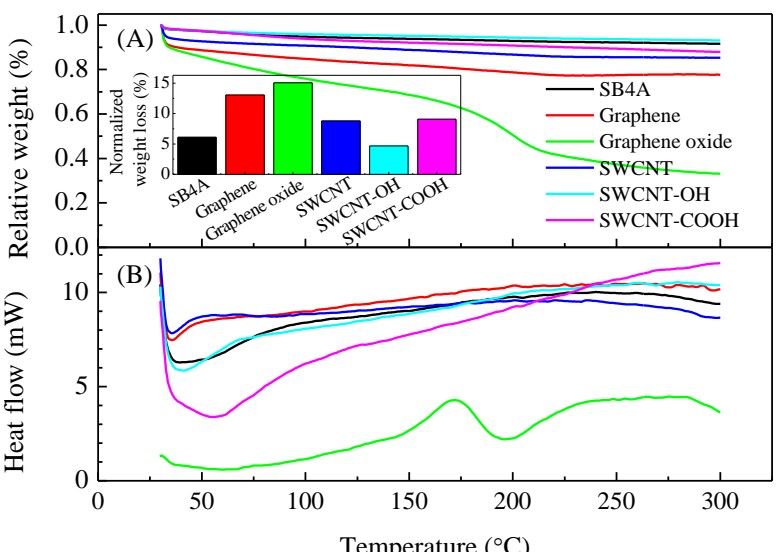


Fig. 3.

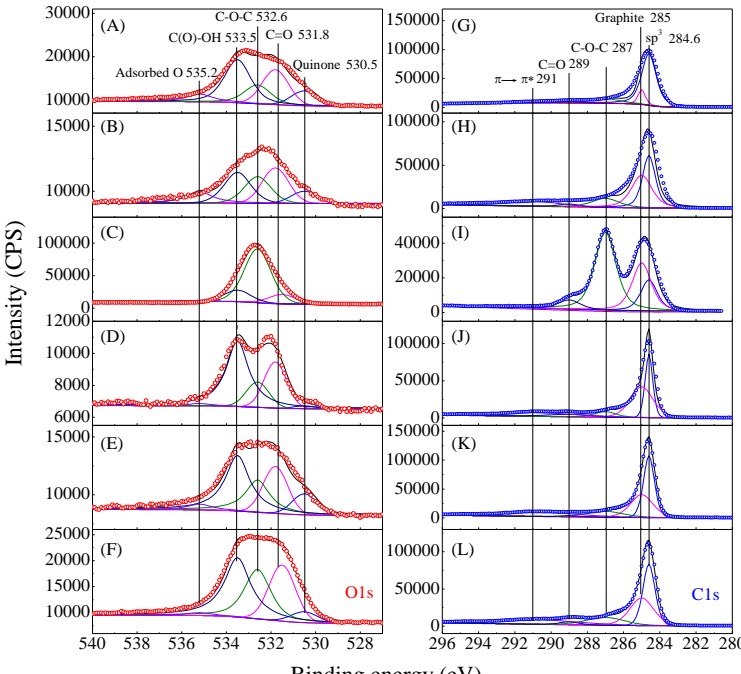






Fig. 4.

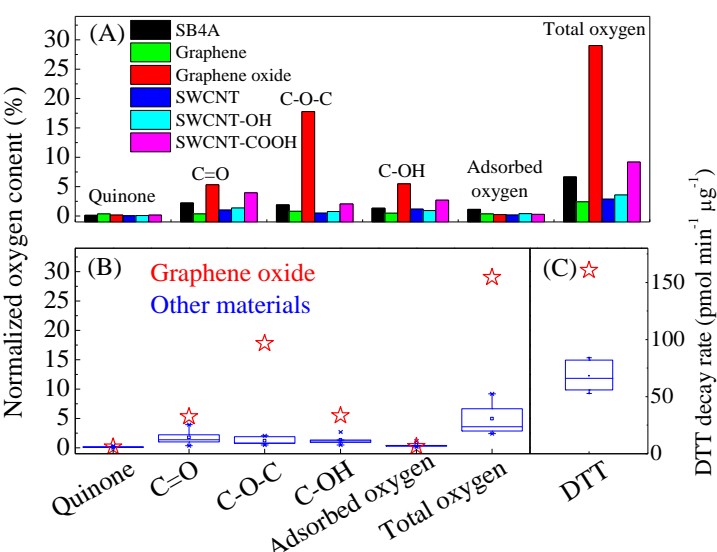


Fig. 5.

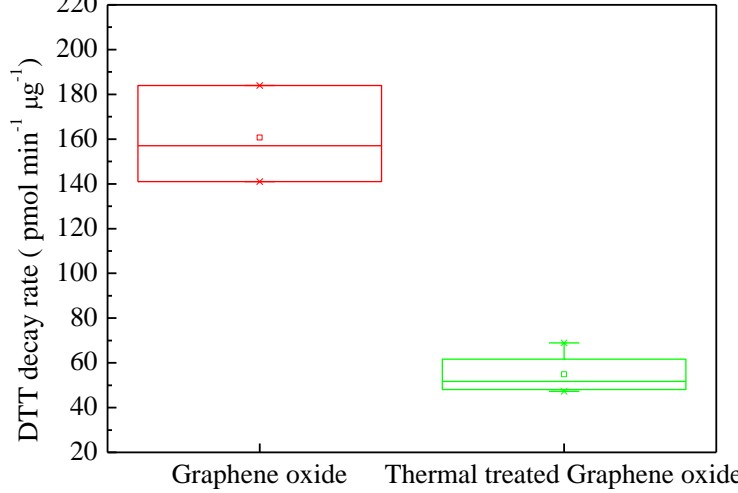


Figure 6.
