# Peer review of "Influence of functional groups on toxicity of carbon nanomaterials: implication for toxicological evolution during atmospheric relevant aging of soot"

_Atmospheric Chemistry and Physics, 2018_

## Referee Comment (RC1) · Anonymous Referee #1 · 13 Mar 2019

General Comments: This work deals with the very complex nature of how small particles affect human health. There is substantial evidence that shows that small particles do have adverse health effects, but it is still not clear what causes issues. The authors recognize this, and do a good job of presenting the problem and previous research (with one notable exception, which will be discussed below). They focus on different carbon-based nanoparticles, including engineered ones. Their results are generally in agreement with previous work, but their major new contribution is the identification of epoxide groups on graphene oxide surfaces having a significantly larger effect on DTT

decay rates.

I have no major issues with how the experiments were performed (including the analytical methods used), and why the different nanomaterials were used. However, I have some issues with the interpretation and implications stated. The title states "implication for toxicological evolution during atmospheric relevant aging of soot" but most of the results are for engineered nanomaterials, not atmospheric soot, which is a very different type of particle. While it is important to learn about the toxicity of engineered nanomaterials, those results should not be applied to atmospheric soot particles (which have very few engineered nanoparticles). The results themselves are interesting and important enough without the atmospheric extrapolation, and I suggest revising the title and the Conclusions section.

In the discussion section, reasons for the observed effects are given with very little evidence (though there are some references stated). For example, line 248 states: This means the cell membrane might be intact when exposed to SB4A. Another is line 293: For example, adhesions and/or covering on cells could be the main MOA for graphene/graphene oxide (2-D structure), while for carbon nanotubes (1-D structure), piercing and/or internalization by cells could be the main MOA. I suggest moving these types of sentences to the Discussion section and providing more references or information about these assumptions.

Specific Comments:As the authors correctly state, the particles studied have very different chemistry and morphology, making it almost impossible to discern the mechanisms or chemicals responsible for the observed results. While most of the particles have similar DTT decay rates, this could be a coincidence or the result of a similar mode of action. I do not think the authors have clearly identified which it is.

The authors should reference the many works from the Prof. Barry Dellinger group at LSU (he is deceased, but the work continues). They have identified a new type of radical, called Environmentally Persistent Free Radicals, that produce cell damage in a

catalytic cycle involving metals, nanoparticles, and quinones (they have many papers in ES&T). The catalytic cycle upsets the notion that it is simply the concentration of an active species that is important.

Technical Corrections: The paper, in general, is well written. However, there are several awkward sentences, missing or unnecessary words that should be corrected (as a native English speaker, I cannot imagine writing a paper in a different language). These are all minor and did not affect my review.

Some examples are: Line 21 were investigated for understanding - change to "were investigated to understand" Line 63 Change NO3 to NOx Line 94 Functionalized does not to be capitalized Line 348 Do you mean bonded? Line 388 awkward sentence Line 447 add "the" Line 487 don't need this sentence Line 505 explain the difference between BC and CB. They are not the same type of particle
* * *

---

## Author Comment (AC1) · 18 Apr 2019

Referee #1 General Comments: This work deals with the very complex nature of how small particles affect human health. There is substantial evidence that shows that small particles do have adverse health effects, but it is still not clear what causes issues. The authors recognize this, and do a good job of presenting the problem and previous research (with one notable exception, which will be discussed below). They focus on different carbon-based nanoparticles, including engineered ones. Their results are generally in agreement with previous work, but their major new contribution is

the identification of epoxide groups on graphene oxide surfaces having a significantly larger effect on DTT decay rates.

Response: Thank you for your positive comments.

I have no major issues with how the experiments were performed (including the analytical methods used), and why the different nanomaterials were used. However, I have some issues with the interpretation and implications stated. The title states "implication for toxicological evolution during atmospheric relevant aging of soot" but most of the results are for engineered nanomaterials, not atmospheric soot, which is a very different type of particle. While it is important to learn about the toxicity of engineered nanomaterials, those results should not be applied to atmospheric soot particles (which have very few engineered nanoparticles). The results themselves are interesting and important enough without the atmospheric extrapolation, and I suggest revising the title and the Conclusions section.

Response: Thank you for your instructive suggestions. We revised the title as "Influence of functional groups on toxicity of carbon nanomaterials".

In the Conclusion section, we removed the sentences related to soot aging. For example, the following sentences have been deleted in the revised manuscript. "It is also a primary process in the atmosphere relating to chemical aging of particles including soot and CB particles", "This means that oxidation potential enhancement of CB particles is also possibly resulted from the formation of epoxide during chemical aging in the atmosphere", "On the other hand, it has been found that aging rate of BC particles under highly polluted urban environment is faster than that under clean conditions (Peng et al., 2016). In the future, much work should be performed on the toxicity evolution of CB or BC particles under real atmospheric conditions".

In the abstract section, the final sentence has been revised as "These results imply that epoxidation might enhance the oxidation potential of carbon nanomaterials". In the introduction section, the sentence "In the current study, both the cell-free toxicity and the cell cytotoxicity of carbon nanomaterials with different functionalities were evaluated to focus on the role of functionalization in their toxicities to understand the possible influence of different source or oxidation processes on the toxicity evolution of soot particles" has been revised as "In the current study, both the cell-free toxicity and the cell cytotoxicity of carbon nanomaterials with different functionalities were evaluated to focus on the role of functionalization in their toxicities".

A new sentence has been added into the conclusion to emphasis the toxicity of epoxide-containing carbon materials as "This means that exposure to epoxide-containing carbon materials should lead to high health risk regarding to oxidation potential" (lines 544-545 in the revised manuscript).

In the discussion section, reasons for the observed effects are given with very little evidence (though there are some references stated). For example, line 248 states: This means the cell membrane might be intact when exposed to SB4A. Another is line 293: For example, adhesions and/or covering on cells could be the main MOA for graphene/graphene oxide (2-D structure), while for carbon nanotubes (1-D structure), piercing and/or internalization by cells could be the main MOA. I suggest moving these types of sentences to the Discussion section and providing more references or information about these assumptions.

Response: Thank you for your suggestion. In the original manuscript, we did not separate the discussion from the results section. In the revised manuscript, we devided them into two parts. These sentences you mentioned have been moved to discussion section as "As shown in Fig. 2, all the carbon nanomaterials showed decreased ATP activities as a function of the dose. This means the carbon nanomaterials investigated in this work are toxic to murine J774 cell line. This is consistent with the previous results that CNT and Printex U are toxic to J774 cells (Kumarathasan et al., 2012) and graphene oxide can induce dose-dependent cell death in normal lung fibroblasts (HLF), macrophages (THP-1 and J744A), epithelial (BEAS-2B) cells, lung cancer cells (A549) etc. (Zhang et al., 2016;Li et al., 2018). At the same time, the BrdU activities decreased as a function of the dose of carbon nanomaterials, which means they are inhibitor for cell proliferation of murine J744 (Cappella et al., 2015). In addition, except for SB4A, other carbon nanomaterials showed significant increases in LDH. This means that the integrity of cell membrane decreased when J774 cells were exposed to these engineered carbon nanomaterials, while the cell membrane might be intact when exposed to SB4A (Cho et al., 2008;Kumarathasan et al., 2015). This might be related to lipid peroxidation induced by these engineered particles (Li et al., 2018) and the non-sphere feature of these engineered particles as observed in Fig.S1. These results also consistent with the previous study that observed CNT cytotoxicity ranking was assay-dependent (Kumarathasan et al., 2015)". The corresponding references have been added in the revised manuscript as you suggested.

Specific Comments: As the authors correctly state, the particles studied have very different chemistry and morphology, making it almost impossible to discern the mechanisms or chemicals responsible for the observed results. While most of the particles have similar DTT decay rates, this could be a coincidence or the result of a similar mode of action. I do not think the authors have clearly identified which it is. The authors should reference the many works from the Prof. Barry Dellinger group at LSU (he is deceased, but the work continues). They have identified a new type of radical, called Environmentally Persistent Free Radicals, that produce cell damage in a catalytic cycle involving metals, nanoparticles, and quinones (they have many papers in ES&T). The catalytic cycle upsets the notion that it is simply the concentration of an active species that is important.

Response: Thank you for your instructive suggestion. We agree with you that the observed DTT activity could be a coincidence of the chemical composition, functional groups and morphology of these particles. This make it is difficult to clearly identify the crucial factor determining the toxicity. This is the reason why we mainly compared the toxicity among these particles with the similar morphology, in particular, between graphene and graphene oxide. On the other hand, we think the role of epoxide in the highest DTT activity of graphene oxide can be well supported by the different DTT activity between thermal treated graphene oxide and the pristine graphene oxide. In the revised manuscript, we emphasized this point as "Although the observed toxicity including DTT activity and cytotoxicity could be a coincidence of the chemical composition, functional groups and morphology of these particles, the above results at least imply that these physiochemical properties such as morphology, metal and OC content should not be crucial factors as for the toxicity of these carbon nanomaterials because it is difficult to observe an obvious dependence of the toxicity on these factors" (lines 483-488 in the revised manuscript).

As for the Environmentally Persistent Free Radicals, we added a paragraph in the revised manuscript (lines 514-520) as "Recently, environmentally persistent free radicals (EPFRs) (a kind of surface stabilized metal-radical complexes characterized by an oxygen-centered radical) (Dugas et al., 2016) have been identified in different source of particles including biomass/coal combustion, diesel and gasoline exhaust, ambient PM2.5 and polymer (Balakrishna et al., 2009;Truong et al., 2010;Dugas et al., 2016). However, it is unclear that whether epoxide in graphene oxide observed in this study contributes to the EPFRs formation. This is needed to be investigated in the future".

Technical Corrections: The paper, in general, is well written. However, there are several awkward sentences, missing or unnecessary words that should be corrected (as a native English speaker, I cannot imagine writing a paper in a different language). These are all minor and did not affect my review.

Some examples are: Line 21 were investigated for understanding - change to "were investigated to understand"

Line 63 Change NO3 to NOx

Line 94 Functionalized does not to be capitalized

Line 348 Do you mean bonded?

Line 388 awkward sentence

Line 447 add "the"

Line 487 don't need this sentence

Line 505 explain the difference between BC and CB. They are not the same type of particle.

Response: Thank you so much for your comments. We carefully corrected these errors.

Line 21 (line 19 in the revised manuscript): "were investigated for understanding" has been changed to "were investigated to understand".

Line 63 (line 81 in the revised manuscript): "NO3" has been changed to "NOx".

Line 94 (line 112 in the revised manuscript): "fMWCNTs" has been changed to "FMWC-NTs".

Line 348 (line 305 in the revised manuscript): This sentence has been revised as "This can be ascribed to desorption of surface adsorbents including bonded organics and trace water".

Line 388 (line 333-338 in the revised manuscript): This sentence has been revised as "Several oxygen-containing species were observed as shown in Fig. 4A-F. Adsorbed oxygen was observed at 535.2 eV in the O1s spectra. Carbon-oxygen single bond in hydroxyl group (C-OH) and epoxide (C-O-C) were at 533.5 and 532.6 eV, respectively. Carbon-oxygen double bound (C=O) was observed at 531.8 eV, while highly conjugated form of carbonyl oxygen such as quinone groups was identified at 530.5 eV (Schuster et al., 2011)".

Line 447 (line 494 in the revised manuscript): "the" has been added before C-O-C (epoxide).

Line 487: The sentence "The DTT decay rates of special black 4A (SB4A), graphene, graphene oxide, single wall carbon nanotubes (SWCNT), SWCNT-OH and SWCNT-COOH were 45.9ïĆś3.0, 58.5ïĆś6.6, 160.7ïĆś21.7, 38.9ïĆś8.9, 57.0ïĆś7.2 and 36.7ïĆś0.2 pmol min-1ïA▪g-1, respectively. Epoxide has been for the first time identified as a highly active functional group in the carbon nanomaterials as far as the oxidation potential is considered." has been deleted in the revised manuscript.

Line 505: The definition of CB and BC was added in the revised manuscript (lines 36-56) as "Carbon nanomaterials are predominantly composed of carbon atoms, only one kind of element, but they have largely diverse structures characterized by different degrees of crystallinity and different macro- and micromorphology (Somiya, 2013). Their basic structure is that of graphite with planes of honeycomb-arranged carbon atoms. Carbon black (CB), which is produced from incomplete combustion of heavy petroleum materials under controlled conditions (Apicella et al., 2003), has been widely used in industrial products, such as inkjet printer ink, rubber and plastic products (Lee et al., 2016), electrically conductive plastics (Parant et al., 2017), paints, coatings and cosmetics (Sanders and Peeten, 2011) and so on. CB is a quasi-graphitic form of nearly pure element carbon (EC, consist of graphene layers). It is distinguished by its very low quantities of extractable organic compounds and total inorganics (Long et al., 2013) compared with soot or black carbon (BC) (Andreae and Gelencser, 2006). Soot or BC, which originates from incomplete combustion of biomasses, biofuels, fossil fuels and natural fires in reduced or anoxic environments, is a mixture of elemental carbon and organic carbon (OC) compounds (Muckenhuber and Grothe, 2006). In addition, as a class of engineering nanoparticles, carbon nanotubes (CNTs) and graphene materials are also a large group of carbon nanomaterials although their graphene sheets are arranged more regularly (Hu et al., 2010) than that in CB (Nienow and Roberts, 2006). During production and use of these consumer products, they are prone to enter into the environment and ultimately the human body (Helland et al., 2007;Tiwari and Marr, 2010), subsequently, to pose risk of adverse health effect".

[revised manuscript text omitted]

                          Fig. 1

[Figure]

                          Fig. 2

[Figure]

                              Fig. 3.

[Figure]

Fig. 4.

[Figure]

Fig. 5.

[Figure]

Figure 6.

---

## Referee Comment (RC2) · Anonymous Referee #2 · 19 Apr 2019

In this work, the authors study the DTT and cytotoxicity response of several carbon nanomaterials and correlate them to their morphology and chemical composition. The main finding is that the epoxide content of graphene oxide is particularly high and also results in high apparent oxidative potential. This specificity is confirmed with thermal treatment of this substance to reduce the epoxide abundance (though also accompanied by morphological changes in the process). The manuscript is generally well-written and addresses a current topic to interest of many researchers. The measurements appear technically sound, though further comments below could be addressed

to make the manuscript suitable for publication.

General comments.

First question is regarding the XPS measurements:

* How do the authors go from counts per second to oxygen content in (%) in Figure 5? If no calibration is performed, then is it possible to state absolute differences among functional groups or only C-O-C content among different materials?

* How are epoxides distinguished from ethers?

* It's not clear that these functional group characterizations are representative of the overall OC that is separately measured given the small probing depth of XPS. Can the authors comment on this?

The oxidation of SO2 by epoxides 2016 is cited as support for ROS generation observed in this work, but the cited work of He and He (2016) proposes a surface binding mechanism that is different from the mechanism by which oxidative potential of ROS is meant to be measured by DTT. The authors may wish to clarify this point as this may also be related to the discrepancy with the lack of difference in apparent cytotoxicity.

As with the other reviewer I agree that the connection to atmospheric soot particles is quite tenuous; due to my delay in response I already see that the authors have proposed changes in this regard (which makes the work less relevant for ACP?). One additional point on this is that the authors refer to "BC" but perhaps "soot" is more suitable, and the "surface functionalization" of soot have been characterized previously (including ethers) - e.g., Cain et al. 2010, Vander Wal et al. 2011, Popovicheva et al. 2014. However, atmospheric aging not includes surface functionalization but also condensation of co-emitted species and photochemical oxidation products which are particularly rapid under conditions of soot emissions (Johnson et al. 2005 and Adachi et al. 2010); it is unclear how much of the oxidation potential attributable to functional groups would be dependent on the carbon nanomaterial itself in the environmental

context.

Cain, J. P.; Gassman, P. L.; Wang, H. & Laskin, A. Micro-FTIR study of soot chemical composition-evidence of aliphatic hydrocarbons on nascent soot surfaces, Physical Chemistry Chemical Physics, 2010, 12, 5206-5218

Johnson, K. S.; Zuberi, B.; Molina, L. T.; Molina, M. J.; Iedema, M. J.; Cowin, J. P.; Gaspar, D. J.; Wang, C. & Laskin, A. Processing of soot in an urban environment: case study from the Mexico City Metropolitan Area, Atmospheric Chemistry and Physics, 2005, 5, 3033-3043

Adachi, K.; Chung, S. H. & Buseck, P. R. Shapes of soot aerosol particles and implications for their effects on climate Journal of Geophysical Research-atmospheres, 2010, 115, D15206

Popovicheva, O. B.; Kireeva, E. D.; Shonija, N. K.; Vojtisek-Lom, M. & Schwarz, J. FTIR analysis of surface functionalities on particulate matter produced by off-road diesel engines operating on diesel and biofuel, Environmental Science and Pollution Research, 2014, 22, 4534-4544

Vander Wal, R. L.; Bryg, V. M. & Hays, M. D. XPS Analysis of Combustion Aerosols for Chemical Composition, Surface Chemistry, and Carbon Chemical State, Analytical Chemistry, 2011, 83, 1924-1930

Minor comments:

The methods section is very sparse in citations except a few of the authors own work, but citations to primary sources would be relevant here.

There are typographical and grammatical errors which can be corrected during the editorial process of Copernicus.

---

## Author Comment (AC2) · 25 Apr 2019

Referee #2

In this work, the authors study the DTT and cytotoxicity response of several carbon nanomaterials and correlate them to their morphology and chemical composition. The main finding is that the epoxide content of graphene oxide is particularly high and also results in high apparent oxidative potential. This specificity is confirmed with thermal treatment of this substance to reduce the epoxide abundance (though also accompa-

nied by morphological changes in the process). The manuscript is generally well written and addresses a current topic to interest of many researchers. The measurements appear technically sound, though further comments below could be addressed.

Response: Thank you for your positive comments.

General comments.

First question is regarding the XPS measurements: 1) How do the authors go from counts per second to oxygen content in (%) in Figure 5? If no calibration is performed, then is it possible to state absolute differences among functional groups or only C-O-C content among different materials? 2) How are epoxides distinguished from ethers? 3) It's not clear that these functional group characterizations are representative of the overall OC that is separately measured given the small probing depth of XPS. Can the authors comment on this?

Response: Thank you for your instructive suggestions. 1) About the XPS measurements, the instrument directly outputs the signal of O1s or C1s in cps, which means the number of electrons that escape from surface of the material being analyzed. When calculating the surface atom contents (%), we scaled the peak areas of each element according to the relative sensitivity factors. However, the relative sensitivity factors for each oxygen-containing species in the envelope of O1s are unavailable at the present time. We simply assumed all these oxygen-containing species in O1s having the same sensitivity factors. We agree with you that this might lead to additional uncertainty, while this method is usually used to calculate the relative content of oxygen-containing species (Chen et al., 2017) and absolute oxygen content of each species (Schuster et al., 2011) when comparing among different samples. Therefore, we calculated the relative fraction of each oxygen-containing species, then converted them into oxygen content.

On the other hand, Wepasnick et al. (2011) measured the surface oxygen-containing species in MWCNTs based on chemical derivation techniques in conjunction with XPS.

The oxygen content of COOH, C=O and C-OH in oxidized MWCNTs were (∼3.0%, ∼1.3% and ∼1.0%), respectively. Using the method based on peak fitting in this work, we calculated the oxygen contents of COOH, C=O and C-OH that had been identified in the MWCNTs after oxidized by 70% HNO3. These values were ∼3.9%, ∼2.0% and ∼1.1% and comparable with those measured with chemical derivation (Wepasnick et al., 2011). Therefore, we think the estimated absolute oxygen content in Fig. 5 should be reliable to semi-quantitatively discuss the influence of oxygen-containing species on the DTT decay rates although we agree with you that this might introduce additional uncertainty. In the revised manuscript (lines 447-453), we added the discussion about the possible uncertainty as "At the present time, the relative sensitivity factors for each oxygen-containing species are unavailable. Similar to the method used in the literatures (Chen et al., 2017;Schuster et al., 2011), we simply assumed all these oxygen-containing species in the envelope of O1s having the same relative sensitivity factors. It should be reliable when semi-quantitatively comparing the contents of oxygen-containing species among different samples although additional uncertainties might be introduced for the calculated oxygen content".

2) If other ethers present in the carbon nanomaterials, it should also contribute to the O1s band which might be closed to that of epoxide. However, at the present time, it has been recognized that oxygen species including epoxide, hydroxyl, carbonyl and carboxylic groups present in graphene layer (Inagaki and Kang, 2014;Hunt et al., 2012). Epoxide should dominate the band at 532.6 eV compared with ethers (Hunt et al., 2012). In particular, the TGA results also supported the high content of epoxide in graphene oxide. For other samples, other ethers might overestimate their contents of epoxide. However, this should not have influence on our conclusion that epoxides are related to the high oxidation potential of graphene oxide. This discussion has been added in the revised manuscript (lines 520-529) as "It should be noted that if other ethers present in the carbon nanomaterials, they should also contribute to the O1s band which might be closed to that of epoxide. However, at the present time, it has been recognized that oxygen-containing species including epoxide, hydroxyl, carbonyl

and carboxylic groups present in graphene layer (Inagaki and Kang, 2014;Hunt et al., 2012). Epoxide should dominate the band at 532.6 eV compared with ethers (Hunt et al., 2012). In particular, the TGA results also supported the high content of epoxide in graphene oxide. For other samples in this work, other ethers might overestimate their contents of epoxide. However, this should not have influence on our conclusion that epoxides are related to the high oxidation potential of graphene oxide".

3) These oxygen-containing species measured using XPS are not representative of the overall OC because the probe depth of XPS is around 10 nm. On the other hand, OC includes not only the oxygen-containing species but also the hydrocarbons without oxygen atoms. Thus, XPS results only reflect the relative element ratio on the surface. However, the surface property should be very important to understand the toxicity of nanoparticles from the point view of particle-cell interaction (Cedervall et al., 2007). In the revised manuscript (lines 343-346), we added sentence to clarify this point as "It should be noted that XPS results only represent the surface atom ratios, which are different from the OC content representing the bulk composition. However, the surface property of particle should be very important to understand the toxicity of nanoparticles from the point view of particle-cell interaction (Cedervall et al., 2007)".

The oxidation of SO2 by epoxides 2016 is cited as support for ROS generation observed mechanism that is different from the mechanism by which oxidative potential of ROS is meant to be measured by DTT. The authors may wish to clarify this point as this may also be related to the discrepancy with the lack of difference in apparent cytotoxicity.

Response: Thank you for your instructive suggestions. We agree with you that the mechanism of SO2 oxidation by epoxide might be different from that of DTT oxidation. Here we cited the oxidation of SO2 by epoxides to support the oxidative property of graphene oxide. In fact, DTT is a stronger reducer than SO2. Both direct oxidation by epoxides and indirect oxidation by ROS generated on the particle surface contribute to the consumption of DTT. Therefore, DTT decay rate should include a part of oxidation reactivity which can be explained by SO2 oxidation. It has also been found that ozone oxidized carbon nanomaterials showed decreased DTT decay rates after treated by SO2 compared with the pristine particles (Xu et al., 2015). In the revised manuscript (lines 517-519), we clarified this point as "This result is also well consistent with the previous founding that epoxides in graphene oxide can oxidize SO2 to sulfate (He and He, 2016) although their oxidation mechanism might be different."

The discrepancy of the observed strong oxidation potential of graphene oxide with the lack of difference in apparent cytotoxicity among these particles may also related to the different interaction mechanism between DTT assay and in vitro assays. In the revised manuscript (lines 400-405), it has been pointed out as "The interaction between target cells and particles should be much complicated than that between DTT and particles. As discussed above, the cytotoxicity of nanoparticles relied on not only the mode of action but also the chemical nature of particles. Therefore, the different responses of the oxidation potential and the cytotoxicity to the epoxide content in these carbon materials might be accounted for by different mechanisms of toxicity among these assays".

As with the other reviewer I agree that the connection to atmospheric soot particles is quite tenuous; due to my delay in response I already see that the authors have proposed changes in this regard (which makes the work less relevant for ACP?).

Response: Thank you for your instructive suggestions. According to your suggestions, we removed the connection to atmospheric soot particles. We think this work is still atmospheric relevant because these carbon nanomaterials can be emitted into the atmosphere from different sources. The results of this work should be still interesting and important enough without the atmospheric extrapolation as commented by reviewer 1. In the revised manuscript (lines 53-56), we added a sentence to emphasize the importance of this work as "During production and use of these consumer products, they are prone to enter into the environment and ultimately the human body (Helland et al., 2007;Tiwari and Marr, 2010), subsequently, to pose risk of adverse health effect".
One additional point on this is that the authors refer to "BC" but perhaps "soot" is more suitable, and the "surface functionalization" of soot have been characterized previously (including ethers) - e.g., Cain et al. 2010, Vander Wal et al. 2011, Popovicheva et al. 2014. However, atmospheric aging not only includes surface functionalization but also condensation of co-emitted species and photochemical oxidation products which are particularly rapid under conditions of soot emissions (Johnson et al. 2005 and Adachi et al. 2010); it is unclear how much of the oxidation potential attributable to functional groups would be dependent on the carbon nanomaterial itself in the environmental context.

Response: Thank you for your instructive suggestions. In the literatures, soot and black carbon are usually exchangeable. In the revised manuscript (lines 47-50), we added a sentence "Soot, which originates from incomplete combustion of biomasses, biofuels, fossil fuels and natural fires in reduced or anoxic environments, is a mixture of elemental carbon and organic carbon (OC) compounds (Muckenhuber and Grothe, 2006)". We replaced the "BC" with "soot" in the revised manuscript according to your suggestion, such as in lines 86, 216, 438 and 549.

The references related to surface functionalization including ethers (Cain et al., 2010;Wal et al., 2011;Popovicheva et al., 2015) have been added in the revised manuscript (lines 97-98).

We agree with you that atmospheric aging not only includes oxidation but also condensation or coating of co-emitted species and secondary products from photochemical oxidation. The relative contributions of these two processes in toxicity changes of soot or CB particles to the oxidation potential are unclear at present time and might be dependent on the carbon nanomaterial. It has been found that the DTT decay rates of SWCNTs (Liu et al., 2015) and engineered nanoparticles ($SiO_2$) (Liu et al., 2019) decreased significantly as a function of exposure time of these pollutants. In the revised manuscript (lines 582-591), we discussed the uncertainty of this work as "Finally, condensation of co-emitted species and photo oxidation products is particularly rapid

under conditions of soot emissions (Johnson et al., 2005;Adachi et al., 2010;Peng et al., 2016). In previous our work, it has been found that condensation process significantly decreased the oxidation potential of the SWCNTs (Liu et al., 2015). A recent work has also found that condensation of organic aerosol leads to decrease in oxidation potential on engineered nanoparticles (Liu et al., 2019). Therefore, the contribution of functional groups to the oxidation potential should be greatly influenced by condensation of co-emitted species and photo oxidation products in the atmosphere. This might be dependent on the carbon nanomaterial itself and needs to be investigated in the future".

Minor comments:

The methods section is very sparse in citations except a few of the authors own work, but citations to primary sources would be relevant here.

Response: Thank you for your instructive suggestions. Several relevant references have been cited in the revised manuscript (lines: 143, 153-154, 168).

There are typographical and grammatical errors which can be corrected during the editorial process of Copernicus.

Response: Thank you for your suggestions. We carefully checked and corrected some typographical and grammatical errors.

[Figure]

**Supplement:**

[revised manuscript text omitted]

                                    Fig. 1

[Figure]

                                    Fig. 2

[Figure]

Fig. 3.

[Figure]

                          Fig. 4.

[Figure]

                          Fig. 5.

[Figure]

                        Figure 6.